# Cell fixation improves performance of in situ crosslinking mass spectrometry while preserving cellular ultrastructure

Andrew R. M. Michael [1,3], Bruno C. Amaral[1,3], Kallie L. Ball[1], Kristen H. Eiriksson [1] & David C. Schriemer [1,2] ✉

Crosslinking mass spectrometry (XL-MS) has the potential to map the interactome of the cell with high resolution and depth of coverage. However, current in vivo XL-MS methods are hampered by crosslinkers that demonstrate low cell permeability and require long reaction times. Consequently, interactome sampling is not high and long incubation times can distort the cell, bringing into question the validity any protein interactions identified by the method. We address these issues with a fast formaldehyde-based fixation method applied prior to the introduction of secondary crosslinkers. Using human A549 cells and a range of reagents, we show that 4% formaldehyde fixation with membrane permeabilization preserves cellular ultrastructure and simultaneously improves reaction conditions for in situ XL-MS. Protein labeling yields can be increased even for nominally membrane-permeable reagents, and surprisingly, high-concentration formaldehyde does not compete with conventional amine-reactive crosslinking reagents. Prefixation with permeabilization uncouples cellular dynamics from crosslinker dynamics, enhancing control over crosslinking yield and permitting the use of any chemical crosslinker.

Maps of protein-protein interactions (PPIs) generate networks that help us understand how cells function and respond to various stimuli. For 20 years or more, these networks have been sampled using affinity pulldown mass spectrometry (AP-MS) experiments[1–3]. However, AP-MS only generates indirect interaction data, and the experiments are prone to false positives. Crosslinking mass spectrometry (XL-MS) could produce higher-quality interaction data by providing a distance measurement between two points in cellular space[4]. It is used very successfully for protein structure determination of reconstituted multiprotein complexes[5–7], thus it should be useful for interactome mapping as well. Ideally, crosslinks would sample the undisturbed spatial proteome. In situ XL-MS experiments have already supported some compelling modeling and interaction studies[8–10], suggesting that in-depth PPI mapping may soon be possible.

Unfortunately, the spatial proteome is not sampled very deeply by conventional XL-MS methods, although some progress is being made[11–13]. After crosslink installation, cells are lysed and digested in a standard bottom-up workflow common to conventional proteomics routines, generating linked peptides for detection by MS. Crosslinked peptides are minor reaction products and cannot be identified with the conventional database searching strategies used in the proteomics community. New software makes the detection of crosslinks more efficient than ever before[14–18], but very little progress has been made in improving the crosslinking reaction itself. To faithfully sample both the spatial and temporal properties of the proteome, crosslinkers must cross the cell membrane, perfuse freely throughout the cell, and react quickly. The crosslinkers most readily available to the community target lysines through N-hydroxysuccinimide (NHS) esters, but the majority of these reagents do not permeate the membrane very well at all[19]. As a result, incubation times as long as an hour are needed to integrate a detectable level of reaction product. The structural proteome could very well be distorted during this timeframe, as cells

[1]Department of Biochemistry and Molecular Biology, University of Calgary, Calgary, Alberta T2N-4N1, Canada. [2]Department of Chemistry, University of Calgary, Calgary, Alberta T2N-4N1, Canada. [3]These authors contributed equally: Andrew R. M. Michael, Bruno C. Amaral. ✉e-mail: dschriem@ucalgary.ca

are dynamic and respond to chemical stimuli. These issues may be addressed by increasing the biocompatibility of the reagents[19,20], but the design constraints are high: maximum crosslink yields must be realized in as short a time as possible.

New faster-acting reagents are not the only way to approach the problem. It would be ideal to stabilize the spatial proteome first: arresting all protein movement and uncoupling cellular dynamics from crosslinker reaction dynamics. Stabilization would permit longer crosslinker reaction times and provide more control over crosslinking yield. If done quickly, it could even capture flux in the interactome. Formaldehyde-based fixation is a compelling stabilization option. Formaldehyde has been used for over 100 years to fix cells for microscopy and preserve biological samples for long-term tissue storage[21,22]. It too is a crosslinker, but it is quite different from the reagents used in XL-MS. Formaldehyde is very water soluble, is rapidly taken up into cells, and its reaction kinetics are fast[23,24]. The mechanism of stabilization is complex. At a common fixative concentration of 4% (1.3 M), formaldehyde exists in equilibrium between reactive free formaldehyde and unreactive methylene glycol, the latter in large excess[25,26]. Fixation appears to involve an initial

burst phase (seconds to minutes) where formaldehyde crosslinks different classes of biological macromolecules, including proteins[23]. Long-term stabilization is thought to involve the slow conversion of methylene glycol to reactive formaldehyde in a type of clock reaction[27]. The burst phase provides the stabilization needed for methods like immunofluorescence that are used to explore cellular mechanisms, whereas the extended incubations enhance long-term stabilization and support tissue preservation for biobanks. Early tests of formaldehyde as a reagent for XL-MS tried to leverage the initial burst phase but were generally unsuccessful[28,29]. A recent study uncovered a double methylene bridge between lysines that could exploited, but the yields from such reactions appear much lower than NHS-based crosslinkers[30]. In addition, the apparent nonselective nature of formaldehyde crosslinking would complicate an already challenging database search space[31].

While it is not yet a viable crosslinker for in vivo XL-MS, it is clearly an effective stabilizer of protein and cellular structures. Very minimal formaldehyde crosslinking is already used to slightly stabilize protein complexes for many applications, including crosslinking[32,33]. High concentrations of formaldehyde were avoided, presumably over concerns that it would compete with lysine-directed crosslinkers. However, as formaldehyde crosslinks are hard to detect, we reasoned that formaldehyde fixation even at high concentration may not strongly interfere with crosslinking reagents. This may seem counter-intuitive based on our current understanding of formaldehyde chemistry. However, the initial stabilization of cells may only involve a very small fraction of amines directly associated with interfaces, thus leaving ample room for post-fixation crosslinking. Here, we describe an in situ XL-MS method that uses conventional formaldehyde fixation protocols to stabilize cells before the introduction of NHS-based crosslinkers. Surprisingly, high-concentration formaldehyde fixation does not interfere with secondary crosslinking reactions at all, and more importantly, it allows us to develop methods that boost cross-linking yields for in situ work.

## Results and Discussion
### Preserving the spatial proteome for XL-MS
Slow crosslinking reactions can distort the equilibrium state of proteins and prevent accurate modeling of protein structures[34]; thus we first examined if these slow reactions would also distort the spatial proteome. We chose to monitor the dynamic actin cytoskeleton in A549 cells, a human epithelial non-small cell lung cancer cell line. The actin cytoskeleton is an essential component of many cellular processes and thus a good indicator of spatial integrity[35]. We adopted a standard protocol for visualizing cells, involving fast fixation with 4% formaldehyde and staining with CF647-phalloidin. The stain labels filamentous actin (F-actin) by binding at the interface between F-actin subunits (Supplementary Fig. 1A). Images show the striated patterns and absence of puncta that are expected for a healthy and stable cell (Fig. 1A, B). These images demonstrate the speed of fixation achievable with formaldehyde.

We then did an experiment where DSS, a widely used cell-permeable crosslinker, was applied at a typical concentration of 1 mM *before* fixation and image analysis. Significant distortions of the proteome were observed, visibly affecting ~70% of cells. Actin filaments were depolymerized and a large number of puncta were formed (Fig. 1A, B). The crosslinker is normally prepared as a dilution from a DMSO stock, given the limited solubility of the reagent. Thus, we next tested the effect of the vehicle alone (2% DMSO) and observed an even higher level of distortion (Fig. 1A, B). Brightfield images of live cells treated with the vehicle revealed apoptosis in many of the cells (Supplementary Fig. 1B). There was very little cellular distortion when we reversed the process: first fixing the cells with 4% formaldehyde, washing them to remove excess formaldehyde, and then crosslinking with DSS (Fig. 1A, B). Thus, conventional methods for in situ XL-MS

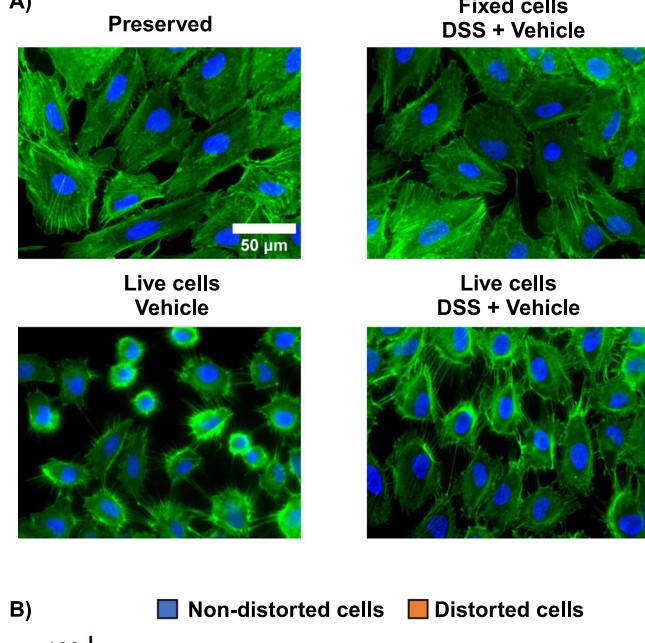

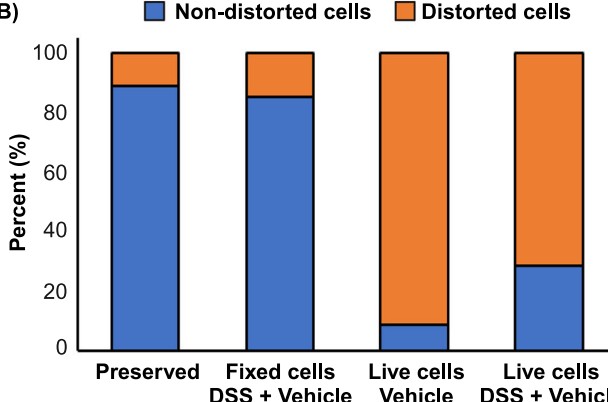

**Fig. 1 | Conventional in situ XL-MS distorts cell structure. A** Fluorescent imaging of actin cytoskeleton (Green) and DNA (blue) in A549 cells for formaldehyde-preserved cells, live cells treated with 1 mM DSS (2% DMSO), live cells treated with 2% DMSO, and formaldehyde-preserved cells treated with 1 mM DSS (2% DMSO). **B** Cells counted for presence (orange)/absence (blue) of cellular disruptions during respective treatments (*n* ≥ 100 cells per treatment). See methods for indicators of disruption. Distorted cells in orange, nondistorted cells in blue. Results consistent with 6 similar experiments.

appear to strongly distort the spatial proteome. Organic solvents are often needed for crosslink reagent solubilization and while the cross-linkers themselves can reduce cellular distortion to a small degree, pre-fixing with a much faster-acting reagent like formaldehyde allows us to separate the stabilization phase from the crosslinking phase.

## Labelling the stabilized spatial proteome

However, we cannot assume that DSS even labels protein after pre-fixation. To explore this, we used *N*-(propionyloxy)succinimide as a surrogate for DSS. Single labeling events are much easier to detect and quantitate than crosslinking events, allowing us to measure a percent labeling of the entire proteome (see methods). Surprisingly, the application of formaldehyde had no effect on the level of reagent incorporation, even up to 4%, and results are independent of cell type (Fig. 2A and Supplementary Figs. 2A, B). Labeling was extensive. Labels were incorporated across the detectable dynamic range of the pro-teome and exhibited no major bias in sampling (Fig. 2C, D and Sup-plementary Fig. 2C). These results are entirely dependent on the washing step, however. Lysine labeling was completely suppressed in the presence of formaldehyde (Supplementary Fig. 2D), consistent with the formation of a methylol derivative and/or a Schiff base[23], which can be reversed by washing the cells before applying DSS. These results indicate that the pool of accessible amines is largely unchanged by formaldehyde-based fixation and suggest that the crosslinking experiment on the fixed cells was successful (Fig. 1).

Before confirming this conclusion, we explored how fixation could improve the permeability of crosslinking reagents, again using monovalent NHS esters as surrogates. It is standard practice in immunofluorescence to use surfactants like Triton-X 100 to per-meabilize cells. These surfactants insert into the lipid bilayer and partially erode the integrity of the membrane. Common surfactant

concentrations for washing-in fluorescent stains and antibody con-jugates are 0.1-0.5%. Here, we fixed A549 cells with 4% formaldehyde and then permeabilized with 0.1% Triton-X 100, on the low end of the scale. We then chose biotin-X-NHS as a surrogate for a larger crosslinker and sulfo-NHS-LC-biotin as a surrogate for a charged crosslinker. Neither reagent labeled the proteome of the non-permeabilized cells. (High levels of apoptosis were observed upon treatment, which can dramatically decrease the pH of the cytosol and reduce the labeling efficiency of NHS esters.) The labeling levels increased substantially upon fixation, presumably because for-maldehyde at this concentration has a mild permeabilization effect[36]. The labeling increased even further with surfactant-based permeabi-lization (Fig. 2B). Fixation with permeabilization supported longer reaction times, even multiple sequential reagent additions, all with no major distortion of the spatial proteome (Supplementary Fig. 3). Taken together, these findings suggest that the full range of chemical crosslinkers could be applied to a pre-stabilized spatial proteome.

## Crosslinking of fixed cells

We then explored how pre-fixation influences the crosslinking reaction itself, using the workflow shown in Fig. 3. This strategy was designed to compare fixed with unfixed reaction conditions and provide a suffi-cient depth of coverage to determine the influence of pre-fixation on relative yields. It was not designed for maximum crosslinker detection, as only 5–12 microgram of total digest was analyzed. We again chose DSS for the comparison. It is a common crosslinker, but one that generates only modest in situ crosslinking yields in conventional XL-MS experiments[37]. Our results confirmed this (Fig. 4A). In a 30 min reaction, less than 30,000 crosslink spectrum matches (CSMs) were detected from live A549 cells, corresponding to 5956 unique crosslinks at a 5% FDR. Most of these crosslinks are loop-links and intra-protein

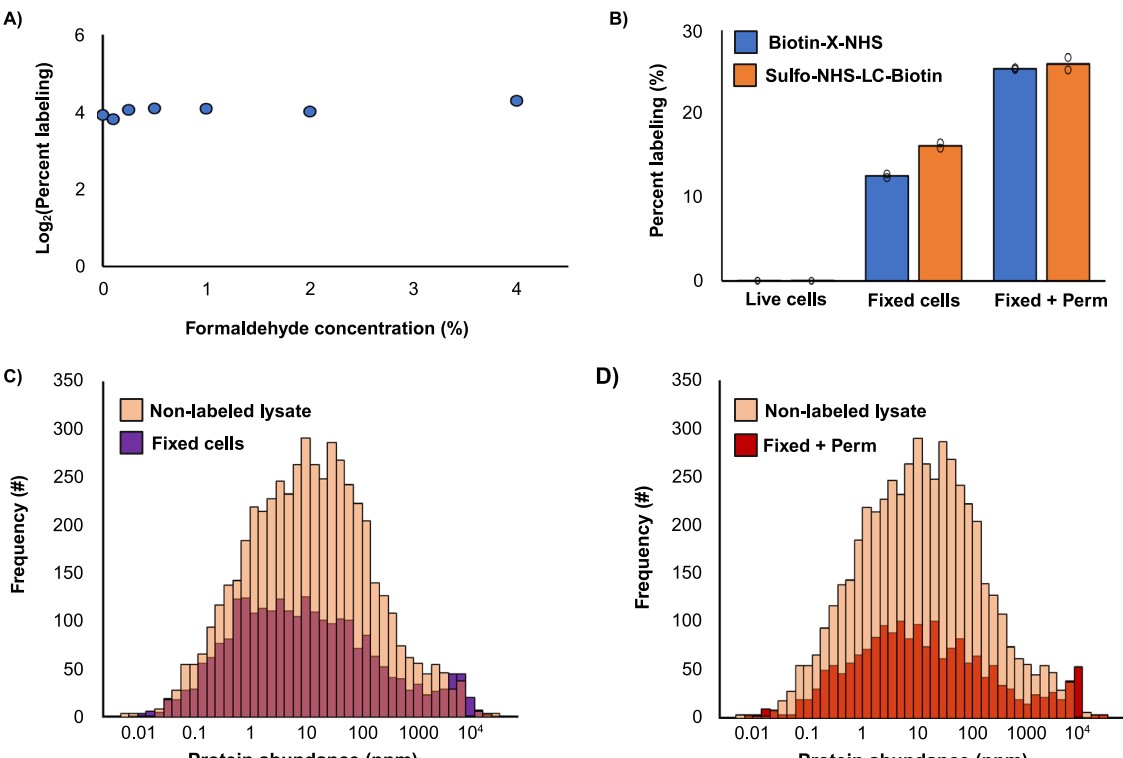

**Fig. 2 | Pre-stabilizing cells with formaldehyde does not impact protein label-ing. A** Average percent proteome labeling using N-(propionyloxy)succinimide across E. coli and human A549 cells with increasing concentrations of for-maldehyde (*n* = 2 biological replicates). **B** Comparison of percent proteome label-ing with biotin-X-NHS (blue) and sulfo-NHS-LC-biotin (orange) in A549 cells for live, fixed, and fixed + permeabilized states (*n* = 2 biological replicates). **C** Histogram of identified protein abundance from human non-labeled lysate (light orange), fixed only (purple), and **D** fixed + permeabilized (red), labeled with biotin-X-NHS. Protein abundancies retrieved from PaxDb[46].

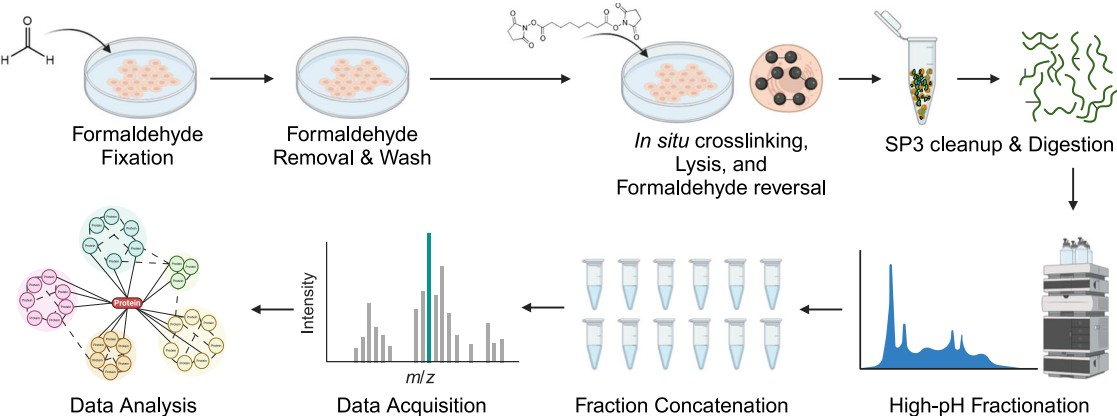

**Fig. 3 | Workflow for formaldehyde pre-stabilization followed by in situ XL-MS.** Cells are fixed with 4% formaldehyde for 10 minutes. After fixation, excess formaldehyde is washed away prior to the introduction of the crosslinker. After secondary crosslinking, cells are collected, lysed, and formaldehyde linkages are reversed by boiling. Extracted protein is then cleaned up via a single-pot, solid-phase-enhanced sample-preparation (SP3)[45] protocol and digested overnight with trypsin. Peptides then undergo high-pH fractionation for LC-MS/MS data acquisition. Data were then processed using pLink 2[17]. Created with Biorender.com.

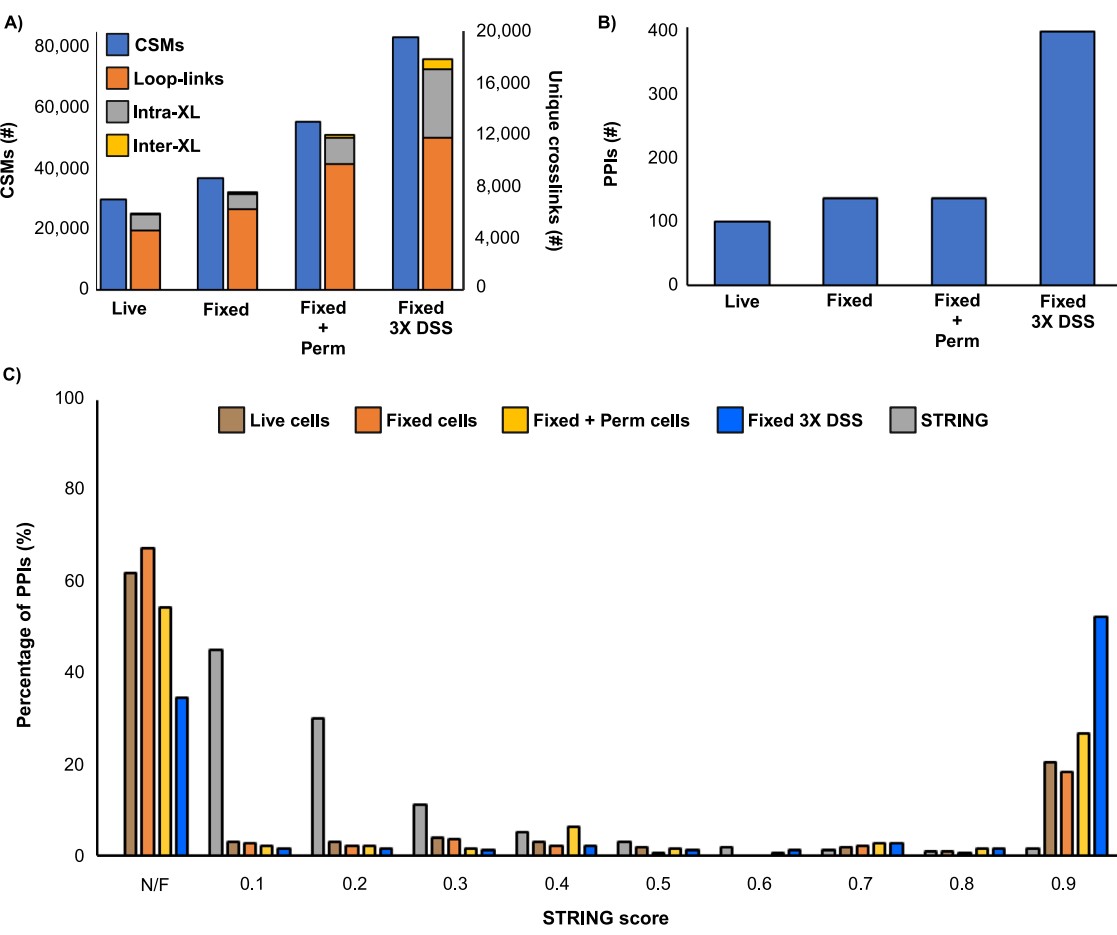

**Fig. 4 | Effect of fixation and permeabilization on in situ XL-MS. A** Number of Number of Crosslink Spectral Matches (CSMs, blue) and unique crosslinks (orange: loop-links, grey: intraprotein, and yellow: interprotein) identified from in situ crosslinking with DSS, in either live cells, fixed cells, fixed and permeabilized cells, or fixed cells with a 3X DSS treatment. **B** Number of Protein-Protein Interactions (PPIs) identified from in situ crosslinking with DSS, in either live cells, fixed cells, fixed and permeabilized cells, and fixed cells with a 3X DSS treatment. **C** STRING score distribution for STRING PPIs (grey), live cells (brown), fixed cells (orange), fixed + permeabilized cells (yellow), and fixed cells with a 3X DSS treatment (blue); PPIs not found in STRING database labeled as not-found (N/F).

linkages as expected, and we observed ratios of reaction products similar to those generated for free proteins and complexes[38]. (See Supplementary Data 1 for a complete breakdown of reaction products for 5% and 1% FDR.)

Fixation alone was insufficient to enhance yields for DSS, even though it increased the labeling yield of monomeric labeling agents. Compared to live cells, fixation only led to a 1.2-fold increase in CSMs and no change in the number of unique crosslinks (Fig. 4A). However,

after treating fixed cells with 0.1% Triton-X 100, the number of CSMs and unique crosslinked peptides increased almost two-fold (Fig. 4A). The overall quality of these identifications was superior to the standard analysis in both yield and score, and similar ratios of reaction products were observed (Supplementary Fig. 4 and Supplementary Data 1). These conditions generated a modest 136 PPIs, (Fig. 4B), so we attempted to increase yield through multiple applications of DSS. We chose three sequential 1 mM treatments with DSS based on our observation that a 3X application retained ultrastructure (Supplementary Fig. 3C). Multiple treatments resulted in a three-fold increase in CSMs and a four-fold increase in unique crosslinks, compared to live cells (Fig. 4A). The yield of PPIs also increased ~3-fold to 399 (Fig. 4B and Supplementary Data 1, 2). We observed that multiple treatments also improved the quality of the detected PPIs: STRING scores were 0.9 for most hits (Fig. 4C). Most of the PPIs that we detected should be well represented in existing databases at this level of interactome sampling, as our fractionation strategy is biased towards more abundant (and hence better studied) proteins. Taken together, fixation with permeabilization enables a flexible crosslinking protocol where yields can be controlled and amplified where needed.

## Crosslinking of fixed cells—PhoX enriched

Although we can increase the yield of crosslinking reactions significantly over conventional methods, detection will benefit from selective enrichment of the reaction products. One of the most effective strategies involves isolation through an affinity tag in the spacer group situated between the two reactive centers, but there are few such reagents that efficiently cross the membrane. Fixation followed by permeabilization should allow any crosslinker to be used for in situ XL-MS, with perhaps only a slight adjustment of the surfactant concentration. To explore this idea, we tested the PhoX crosslinker in a 1X and 3X treatment. This reagent contains a negatively charged phosphonate that can be enriched by immobilized metal ion affinity chromatography (IMAC) resins[39], using a very simple process that is common in proteomics. However, the negatively charged phosphonate renders the molecule membrane-impermeable, restricting the crosslinker to lysates mostly. Derivatized versions of PhoX have been developed recently with improved penetrance[20], but our goal was to explore if completely impermeable reagents could be introduced.

Fixation with permeabilization enabled an effective in situ reaction. We recovered approximately 13% (wt. percent) of the total peptide digest in this experiment, of which 53% were crosslinked peptides of all types, 8% were verified phosphopeptides, and 39% were putative phosphopeptides and/or free peptides. These enrichment statistics are consistent with a previous crosslinking study[39]. We detected 168,177 CSMs in this data set, which reduced to 32,084 unique crosslinked peptides at a 5% FDR (Fig. 5A and Supplementary Data 1). Yields are approximately two times greater than the 3X DSS crosslinking experiment in both categories (Fig. 4A), reducing to 438 PPIs at a 5% FDR. As with the 3X DSS reaction, most of the detected PPIs were matched to the STRING database with high scores (Fig. 5B, Supplementary Fig. 5, and Supplementary Data 3). Repeating the experiment with a 3X application of PhoX increased crosslinked peptide recovery to 89%. The number of CSMs rose to 299,209, which reduced to 41,812 unique crosslinked peptides at a 5% FDR and 915 PPIs (Fig. 5A). We note that only two 10-cm plates of cells were used as input for this analysis, much less than other in situ crosslinking studies.

A more in-depth analysis of the interprotein crosslinks from the 3X PhoX experiment showed a set of complexes that are consistent with the MS sampling depth that we applied (~5 microgram over 12 fractions) (Fig. 5C, Supplementary Data 4). That is, most protein interactions that we detect involve relatively high abundance proteins, including subunits of the TRiC/CCT molecular chaperone, the 26S proteasome particle, the ribosome, histones, and the DNA

replisome (Fig. 5D). However, additional complexes comprised of proteins with lower copy numbers are also in evidence, including the Ku70/80 heterodimer (Fig. 5E) involved in DNA damage repair, histone regulatory complexes such as histone deacetylase 2 interacting with REST corepressor 1, and the serine/threonine kinase Nek1 interacting with histone 2 A. The latter is most interesting as it highlights that fast fixation with secondary crosslinking can capture transient enzyme-substrate complexes. Mapping the unique crosslinks to known protein and protein complexes generated a histogram of distances that reflect a good sampling of structure: 68% of crosslinks were within 20 Å and 90% of crosslinks within 35 Å for 3X PhoX treatment (Fig. 5F). These numbers were nearly identical to those generated by 1X PhoX treatment (Supplementary Fig. 5). Few overlength crosslinks were observed.

The depth of sampling was sufficient to begin exploring interactors that possess little available structural information. For example, in both the 1X and 3X PhoX crosslinking experiments, we detected an interaction between apoptosis inhibitor 5 (API5) and the apoptotic chromatin condensation inducer in the nucleus (ACIN1), a nuclear complex that regulates apoptotic DNA fragmentation[40]. In our data, the complex was mapped with 2 interprotein crosslinks and 15 intra-protein crosslinks. We generated structures of the dimer using AlphaFold2 multimer[41,42] and found that 53% of crosslinks were within 20 Å and 78% of crosslinks within 35 Å (Supplementary Fig. 6). In all models, interlinks were mapped at distances below 20 Å, whereas several of the overlength intraprotein crosslinks span nominally disordered regions. This example highlights the potential benefit of in situ crosslinking as these linkages could be used to drive a more authentic modeling effort.

In summary, standard in vivo crosslinking protocols can distort the spatial proteome and should be used with caution, although the degree to which spurious PPIs are generated is currently unknown. We show that pre-fixing cells is effective at stabilizing the proteome for in situ XL-MS experiments. Together with permeabilization routines, pre-fixing supports extended reaction times, higher crosslinking yields, and widens the scope of crosslinkers that can be used. It is somewhat surprising that a formaldehyde-based pre-stabilization method would support XL-MS at all, especially at the elevated concentrations we used (4%). Formaldehyde fixation involves the irreversible conjugation of free amines, at least based on many classical bioconjugation texts[43]. However, we demonstrate that most of the modifications arising from a typical fixation experiment are reversible simply by washing the treated cells with buffer or through competition with NHS esters. More recent literature has demonstrated that even terminal lysine-specific reaction products have a range of stabilities. For example, hydroxymethylated and bridged amines are the major products from short-term fixation, which have been shown to be reversible in NMR studies of free amino acids, whereas N-methylation and N-formylation are not[31]. (We did detect N-methylation and N-formylation under extended fixation times.) How then does fixation occur? It would seem to be the product of a broad reaction profile involving multiple different protein residues, and perhaps other biomolecules[23]. Fixation may involve the formation of lysine-mediated crosslinks through an imine-based intermediate, but if so, these can be reversed through treatment with NHS esters. The lability of imines and their reaction product suggest this is possible.

Rapid pre-fixation of cells restores flexibility to method development for in situ XL-MS. Reagents can now be designed for effective PPI mapping without concern over membrane permeability and reaction times. This development will help in situ XL-MS become a viable alternative to AP-MS techniques for interactome analysis. This method should also promote confidence in associations detected by in situ XL-MS because it is based upon an established microscopy technique for mapping the spatial distribution of proteins,

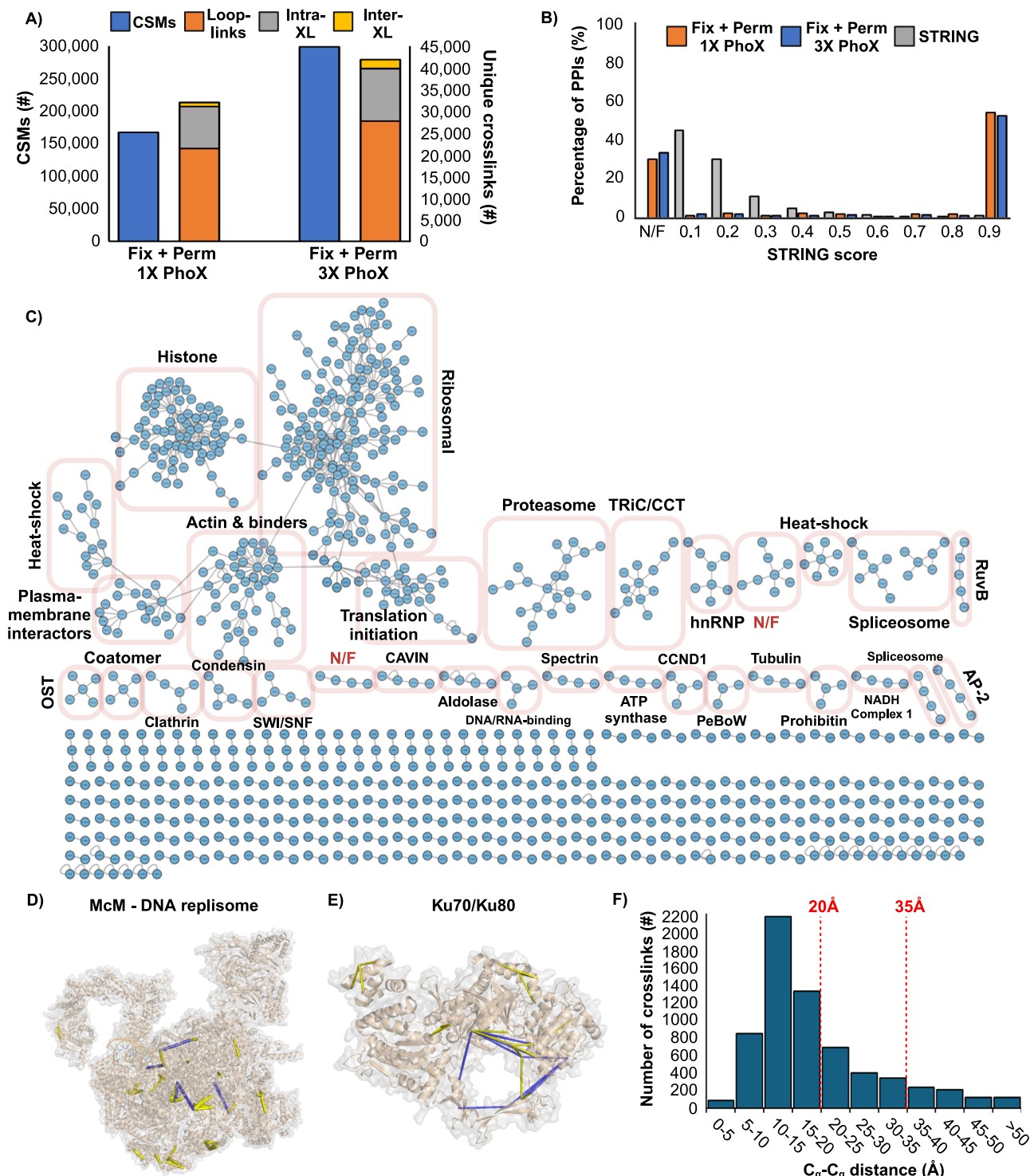

**Fig. 5 | In situ crosslinking of fixed and permeabilized A549 cells with PhoX.**
**A** Number of Crosslink Spectral Matches (CSMs, blue) and unique crosslinks (orange: loop-links, grey: intraprotein, and yellow: interprotein) identified from in situ crosslinking with 1X or 3X PhoX, in fixed and permeabilized cells. **B** STRING score distribution for STRING Protein-Protein Interactions (PPIs, grey), 1X in situ PhoX crosslinked cells (orange), and 3X in situ PhoX crosslinked cells (blue); PPIs not present in STRING database labeled as not-found (N/F). **C** PPI network plot of all detected interactions from 3X in situ PhoX crosslinking. **D** 3X PhoX crosslinks mapped to the McM−DNA replisome (mapped to PDB 7PLO) and **E** Ku70/Ku80 (mapped to PDB 1JEQ). Crosslinks below 20 Å are colored yellow, and crosslinks between 20 Å and 35 Å are coloured blue. **F** Histogram of $C_\alpha$-$C_\alpha$ distances of 3X PhoX crosslinks mapped to all known structures.

## Methods

### Cell culture

E. coli (DH5α) cells were seeded into 50 mL of 2YT + Amp100 (Fisher Scientific) medium and grown overnight at 37 °C. A549 cells, a human non-small cell lung carcinoma cell line (American Type Culture Collection (ATCC), catalogue No. CCL-185), were cultured in Ham's F-12K (Kaighn's) medium (Gibco) supplemented with 10% fetal bovine serum (FBS, Gibco) and 1% penicillin-streptomycin (Gibco) in a humidified incubator at 37 °C in 5% $CO_2$. A549 cells were seeded in a 10 cm Petri dish and grown to 80% confluency for crosslinking experiments.

For microscopy, A549 cells were seeded to a 1 cm microscope coverslip and grown to 80% confluency for experiments.

## Cell fixation and permeabilization
E. coli cells were collected and washed in ice-cold 1X PBS, pH 7.4 to remove growth medium. A549 cells had growth medium removed and washed extensively with 1X PBS, pH 7.4. A fresh 4% formaldehyde solution was prepared in PBS by diluting from a 16% methanol-free stock (Thermo Scientific). Cells were treated with formaldehyde for 10 minutes at 25 °C under gentle mixing then washed with PBS. For permeabilization, A549 cells were treated with 0.1% Triton X-100 in PBS, pH 7.4 for 10 minutes at 25 °C and then washed prior to crosslinking.

## Cell imaging and quantification of cell disruption
A549 cells grown on microscope coverslips were washed with PBS and left either in their live-state or fixed with formaldehyde. Cells then underwent treatment with disuccinimidyl suberate (DSS) or vehicle (2% DMSO) as described below. After treatment, cells were washed extensively, and the unfixed samples were then fixed with formaldehyde, for imaging. Samples were permeabilized with 0.5% Triton X-100 for 10 minutes and stained with CF647-phalloidin (1:40 v/v, Biotium) for 1 hour in the dark under gentle mixing. Stained samples were mounted onto microscope slides with EverBrite + DAPI mounting medium (Biotium) and sealed with clear nail polish. Cells were imaged with an AxioObserver inverted microscope using the 40X oil immersion objective and imaged in the 647 nm and DAPI channels using the ZEN microscopy software. Fluorescent phalloidin micrographs were then evaluated for disruptions to the spatial proteome post-treatment. The features we tracked to investigate gross structural perturbations were major striated actin filaments, diffusive actin structures, the presence of filopodia in confluent regions, and rounded cells. Approximately 100 cells were inspected across 6-7 micrographs, for each treatment (validated by two analysts). Images pseudo-coloured, with brightness adjusted for merged composites using ImageJ software.

## In situ crosslinking and labeling
A549 cells (live or fixed, biological duplicates) were washed three times with PBS prior to labeling reactions. Fresh stocks of the monovalent reagents N-(propionyloxy)succinimide ester (PropNHS, Sigma-Aldrich), Biotin-X-NHS (Biotin-NHS, Cayman Chemical), Sulfo-NHS-LC-Biotin (Sulfo-Biotin-NHS, Thermo Fisher Scientific) and crosslinkers DSS (Cayman Chemical) and disuccinimidyl phenyl phosphonic acid (PhoX, Thermo Scientific)) were prepared in anhydrous DMSO (or water for sulfo-biotin-NHS) at 100 mM and 50 mM respectively. Cells were treated with 1 mM of reagent in PBS, pH 7.4 for 30 minutes at 25 °C with gentle mixing. For multiple applications of reagent, cells were treated with 1 mM of reagent in PBS, pH 7.4 for 30 minutes at 25 °C under gentle agitation with washing in between applications using fresh PBS. Reactions were terminated by addition of 100 mM Tris and incubating for 10 minutes. Cells were harvested and pelleted before being suspended in a formaldehyde-reversal lysis buffer containing 500 mM Tris + 150 mM NaCl + 1% Triton X-100 + 2% SDS + 50 mM 1,4-dithiothreitol (DTT) + 1X cOmplete EDTA-free Protease inhibitor cocktail (Sigma-Aldrich), pH 8[44]. Formaldehyde reversal was used to ensure effective digestion. Cells were lysed by boiling at 95 °C with agitation for 60 minutes, followed by protein denaturation in 8 M urea at 55 °C for 30 minutes. Proteins were then alkylated at room temperature with 250 mM 2-chloroacetamide (CAA) for 30 minutes. Samples underwent SP3-protein cleanup[45] and two rounds of tryptic digestion (1:50 enzyme:substrate, wt/wt overnight at 37 °C, followed by 1:100 enzyme:substrate, wt/wt for 4 hours at 37 °C). Trypsin was quenched by addition of formic acid to 0.5%. Peptides were either desalted with C$_{18}$ ZipTips (Millipore Sigma) or Pierce Peptide Desalting

Spin Columns (Thermo Scientific) for monovalent or crosslinked samples, respectively. PhoX-crosslinked peptides were enriched prior to high-pH fractionation using a Fe-NTA magnetic resin (Thermo Scientific). Peptides were resuspended to a concentration of 1 mg/mL in 80% acetonitrile (MeCN) + 0.1% trifluoroacetic acid (TFA). Fe-NTA beads were washed twice with 80% MeCN + 0.1% TFA with 4X bead volume then resuspended in 0.75X bead volume. Peptides were added to beads at a 1:10 (µL beads:µg peptide) and incubated at ambient temperature for 30 minutes under gentle rotation. Beads were then washed three times with 80% MeCN + 0.1% TFA at 4X bead volume, and once with H$_2$O at 4X bead volume. Peptides were eluted from beads twice with 5% ammonium hydroxide at 2X bead volume for two minutes each time.

## High-pH fractionation of crosslinked peptides
Crosslinked peptides were resuspended in mobile phase A (20 mM ammonium formate, pH 10) and were loaded onto an Agilent 1260 infinity II system. Peptides were accumulated onto a ZORBAX RRHD Extended-C$_{18}$ column (80 Å pore size, 2.1 × 150 mm, 1.8 µm particles, Agilent) at 50 °C. Samples were eluted at a flow rate of 0.2 mL/min using a 54 minute multistep gradient from 5% mobile phase B (100% MeCN) for 6 minutes, then 5–45% B for 34 minutes followed by a hold at 45% B for 5 minutes, then a ramp of 45–80% B for 1 minute with a hold at 80% B for 4 minutes and finally a ramp of 80-5% B for 4 minutes. Fractions were collected every 1.2 minutes and concatenated from 48 to 12 fractions following a concatenation scheme of fractions 1 + 13 + 25 + 37, fractions 2 + 14 + 26 + 38,…, and so on.

## LC-MS/MS analysis of monovalent labeled peptides
Monovalent-labeled peptides were resuspended in mobile phase A (0.1% formic acid) and loaded onto a Vanquish Neo HPLC coupled to a nano-ESI source of an Orbitrap Eclipse (ThermoFisher Scientific). Samples were injected onto a 300 µm × 5 mm PepMap Neo Trap Cartridge peptide trap column (C18, 5 µm particle size, 100 Å pore size, ThermoFisher Scientific) and separated on an EASY-Spray 75 µm x 50 cm PepMap HPLC column (C18, 2 µm particle size, 100 Å pore size, ThermoFisher Scientific) at a flow rate of 300 nL/min at 40 °C using a multistep gradient from 2–30% mobile phase B (80% MeCN in 0.1% formic acid) for 75 minutes, 30–45% B for 45 minutes, 45-99% B for 1 minute, and a 10 minute wash at 99% B. A typical data-dependent analysis used a full MS scan of m/z 375–1800, selecting charge states of 2–6+ for fragmentation and scanning in the orbitrap. MS$^1$ and MS$^2$ scan resolutions were 120,000 and 30,000, respectively. For MS$^2$, samples were isolated with a m/z 1.2 window and underwent a normalized collision energy for stepped-HCD fragmentation of 27, 30, and 33%. Maximum injection time was set to 50 and 54 milliseconds for MS$^1$ and MS$^2$, respectively, and dynamic exclusion was set for 30 seconds.

## LC-MS/MS analysis of crosslinked peptides
Fractionated crosslinked peptides were loaded onto a Vanquish Neo HPLC coupled to a nano-ESI source of an Orbitrap Eclipse (ThermoFisher Scientific). Samples were injected onto a 300 µm × 5 mm PepMap Neo Trap Cartridge peptide trap column (C18, 5 µm particle size, 100 Å pore size, ThermoFisher Scientific) and separated on an EASY-Spray 75 µm × 50 cm PepMap HPLC column (C18, 2 µm particle size, 100 Å pore size, ThermoFisher Scientific) at a flow rate of 250 nL/min at 40 °C using a multistep gradient from 2–30% mobile phase B (80% MeCN in 0.1% formic acid) for 160 minutes, 30–45% B for 20 minutes, 45–99% B for 1 minute, and a 10 minute wash at 99% B. A typical data-dependent analysis was used at a full MS scan of m/z 375-1800, selecting charge states of 3–8+ for fragmentation and scanning in the orbitrap. MS$^1$ and MS$^2$ scan resolutions were 120,000 and 30,000, respectively. For MS$^2$, samples were isolated in a m/z 1.2 window and underwent a normalized collision energy for stepped-HCD fragmentation of 27, 30, and 33%. Maximum injection time was set to 50 and

54 milliseconds for MS[1] and MS[2], respectively, and dynamic exclusion was set for 10 seconds.

## Monovalent data analysis

MS data were analyzed on MS FRAGGER (V20.0)[46] using the LFQ-MBR workflow. The search parameters were set as follows: precursor mass tolerance of 10 ppm, 15ppm for fragment mass tolerance, an enzyme for digestion set to trypsin with a maximum of 2 missed cleavages, peptide length set to 6–50 amino acids and a mass range of 500–5000 u. Carbamidomethylation (ΔM = +57.02) on Cystine was set as a fixed modification. Variable modifications were set as follows: oxidation (ΔM = + 15.99) on methionine, pyro-glu from glutamine (ΔM = −17.03 u), acetylation (ΔM = + 42.01 u) on N-terminus. For the respective monovalent reagents, the mass changes were set as follows: PropNHS (ΔM = +56.03 u) on lysine, and NHS-biotin (ΔM = +339.16 u) on lysine. Labelling was quantified as a percentage of total proteome conversion (Eq. 1):

$$\%labeling = 100 \times \sum_{i}^{m} XIC(labeled\ peptides)_i / \sum_{j}^{p} XIC(labeled\ peptides)_j$$

(1)

where extracted ion chromatograms (XICs) for all non-redundant peptide IDs were assessed as peak intensities. Protein abundancies of labeled proteins were collected using the PaxDb 5.0[47] for *Homo sapiens* and compared against the full human proteome or against identified proteins from a non-labeled lysate.

## Crosslink data analysis

Crosslink data of biological duplicate were integrated and analyzed on pLink 2.3.11[17] with the following parameters: minimum peptide length set to 6; maximum peptide length set to 60; maximum of 3 missed cleavages for trypsin; precursor mass tolerance set as 5 ppm; fragment mass tolerance set as 10 ppm; carbamidomethylation of cysteine (57.021 u) set as fixed modification; oxidation of methionine (15.995 u) set as variable modification. Crosslink masses for DSS and PhoX on lysine were set as 138.068 u and 209.971 u, respectively; monolink masses for DSS and PhoX on lysine were set as 156.079 u and 277.982 u, respectively. Data were searched against the full human proteome (retrieved from Uniprot on February 13, 2023). Results are reported at a 1% and 5% FDR set at the PSM level, with FDR calculations for intra-protein and inter-protein crosslinks evaluated separately. We note that extensive optimization of all conditions (i.e., labeling, enrichment and LC-MS) was performed on shotgun analyses of crosslinked samples, prior to conducting duplicate analysis on the fractionated data sets.

## Crosslink mapping to protein complexes and PPI network analysis

PhoX crosslinks were mapped using CLAUDIO[48] across available PDB and AlphaFold structures. CLAUDIO searches were configured using standard parameters apart from setting crosslink sites to the $C_\alpha$ of lysine residues. Predicted models of the structure for the API5-ACIN1 complex were generated using AlphaFold2-multimer[42] on COSMIC2[41]. Crosslinks to the API5-ACIN1 complex were mapped using xiVIEW[46] and exported as a "PyMOL command file" which was then imported into PyMol version 2.5.8 for visualization. PPIs generated from pLink crosslink spectra output for the first considered interaction and were visualized on Cytoscape version 3.10.1. The human PPI database was downloaded from STRING[49] for analysis of PPI confidence using an in-house python script.

## Reporting summary

Further information on research design is available in the Nature Portfolio Reporting Summary linked to this article.

## Data availability

The crosslinking and labeling data generated in this study have been deposited in the PRIDE partner repository[50] with the dataset identifier PXD051075. The Source data for Figs. 1b, 2a-d, 4c and 5b, f are provided in the Supplementary Information/Source Data file. Source data are provided with this paper.

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

## Acknowledgements

This work was funded by the Natural Sciences and Engineering Research Council of Canada Discovery Grants RGPIN 2017-04879 to D.C.S. All figures created and/or assembled with BioRender.com, released under a Creative Commons Attribution-NonCommercial-NoDerivs 4.0 International license.

## Author contributions

D.C.S., A.R.M.M. and B.C.A. conceptualized the project and designed experiments. A.R.M.M. collected all imaging data. A.R.M.M. and B.C.A. collected all mass spectra data, with assistance from K.L.B. and K.H.E. and analyzed all data. D.C.S. and A.R.M.M. generated the first draft of the manuscript and all authors helped finalize the paper.

## Competing interests

The authors declare no competing interests.
