## [Peer Review File · Nature Communications]

Reviewers' Comments:

Reviewer #1:

Remarks to the Author:

This communication describes a strategy to improve crosslinking mass spectrometry by first performing a fast formaldehyde-based fixation and then permeabilizing cells before adding the chemical crosslinker. The formaldehyde crosslinking can then be reversed before digestion and LC-MS analysis. Overall, the authors make a well-supported case that current in-situ XL-MS strategies distort cell structure and the addition of a formaldehyde fixation step can prevent this distortion while still allowing for labeling events. They also show that formaldehyde crosslinking can be paired with membrane permeabilization to allow larger crosslinking reagents to cross the cell membrane. They then apply this workflow using DSS and PhoX, two different crosslinkers.

Overall, the authors do a thorough job of demonstrating this new workflow is effective at increasing in-situ XL-MS efficiency and expanding the crosslinkers available to do in-situ XL-MS with. The claims are well supported, the methodology is solid, and this work should be reproducible. There are a few minor issues outlined below, but once these are addressed this paper is ready for acceptance.

1.The authors should add a key to the data submitted to PRIDE. Even with a logical naming scheme the number of files make it challenging to match results to the experiments.

2.The authors were very thorough in explaining the indicators of cell disruption in the methods. If possible, they should more explicitly state the degree of disruption required to consider a cell distorted.

3.It may be useful to add a citation or rationale as to how the reverse formaldehyde cross-linking conditions were selected.

4.In Figures 2A and 2B, it would be helpful to say the number of replicates performed to get the error bars ideally in the figure caption.

5.In Figure 4A and 4B, the authors should add error bars and state the number of replicates performed. If it is only one replicate they should explicitly say so and acknowledge any limitations that may add to the analysis.

Reviewer #2:

Remarks to the Author:

The remarkable result of this work is that cell fixation avoids the spatial distortion of proteins caused by prolonged exposure to the cross-linking reaction without preserving cellular structures in advance. In addition, formaldehyde fixation does not interfere with labeling and cross-linking reactions when excess formaldehyde is removed.

The results could be of immediate interest. However, the number of PPIs identified is not a very large number to move the field forward. Perhaps it would have been possible to show that the proposed approach has an undeniable advantage over the cross-linking of unfixed cells (by greatly improving the number and quality of PPIs or shortening the time of the cross-linking reaction...) if the hypothesis had also been validated with cross-linkers more suitable for an in-vivo/in-situ experiment, such as cleavable cross-linkers (DSSO, DSBU...), which reduce the search space and increase the identification of possible interactions. Cleavable cross-linkers would also have allowed a direct comparison of the results with recent studies already published in the literature where mainly cleavable cross-linkers were used.

While for enrichable crosslinking it would have been better to use a newer membrane-permeable crosslinking, tBu-PhoX, or to use both PhoX and tBu-PhoX to compare the results, evaluating the coverage improvement on the PPIs identifications when the fixing step is added.

Therefore, expanding the crosslinkers used could have helped strengthen the hypothesis of the study by directly comparing PPI identifications for each crosslinker (permeable and non-permeable, cleavable and non...) when adding the cell fixation step before the crosslinking reaction, as well as for permeabilization.

The conclusions are not straightforward to me. The results clearly show that if formaldehyde is removed prior to the addition of the crosslinker, it does not interfere with the crosslinking reaction. However, the results state that permeabilization after fixation improves the crosslinker yield, but the data show that the approach that provides more PPI and IntersXL identifications is that of 3 sequential shots, but this strategy was not used for PhoX, so it's not clear if more shots could have provided more PPI identifications. Therefore, which strategy is the best cannot be said with certainty. No lysine modification has been found after fixation, but have FA dimer crosslinkers been searched for (Nat Commun 11, 3128 (2020))?

It would be nice to add overlapping PPI analysis. Are all PPIs found in living cells included in Fix+Perm cells treated? If not, are those that differ in live cells due to disrupted cell structure, artifact, or are the 2 approaches complementary?

Some specific mistakes have been identified:

- Error in the text for the total reported XLs of PhoX:

168177  loop + intra + inter, it should have been 53870 for comparing with 10722
10722  intra + inter or 32084 for comparing with 168177.

- Figure 1 could be improved: the order of the bars should follow the order of the microscope images; while Supplementary Figure 3 would look more convincing if the

same quantification as for Figure 1B were present.

- Figure 5C shows redundant PPIs: the double lanes (edges) connecting the same 2 proteins (nodes) are duplicates. Therefore, fewer PPIs were found than declared.
- Figure 5D & E also lacks a description of the crosslinker line colors.
- Figure 4C capture describe as red the bars for live cells, but it is more gold-ish.
- Some inaccuracies and typos were found in the English, including the syntax; please review it.

Some details about repeat injections are missing. No description of subsequent injections: sequential? Was there incubation between one and another? How long did the reaction last? Did the concentration of DMSO in the solution increase by more than 2% with multiple doses of crosslinker? Did this affect crosslinker penetration? Were the cells permeabilized before? This is not clear. It seems NO, so why was it stated that the permeabilization step is a necessary step to improve the in-situ crosslinking reaction? Why not apply it here, also with the multiple shot approach? In my opinion, a more in-depth analysis could have been done.

Reviewer #3:

Remarks to the Author:

In their Study, Michael et al., demonstrate the advantage of a 2-step crosslinking approach using formaldehyde for fixation and stabilization of protein-protein interactions followed by membrane permeabilization and crosslinking with the amine-reactive crosslinkers DSS or PhoX.

They show that fixation preserves the cellular ultrastructure while no competition to the amine-reactive crosslinkers formaldehyde was reported. This clearly improved their results over a classical one-step crosslink approach. Their proof of principle study in A549 cells shows nice proteome coverage and the idea of making more -also non membrane permeable – linkers accessible for “in vivo” crosslinking is great. However, this reviewer is wondering if the idea is new per-se? Two step crosslinking approaches with formaldehyde to maintain cellular integrity were already reported in previous studies and, to give an example, were summarized in a review from the Huang lab back in 2018 (<https://www.ncbi.nlm.nih.gov/pmc/articles/PMC6022837/>). In that review applications to improve structural characterization but also for in vivo studies in combination with the affinity enrichable DSBSO linker are described. In line with the findings of Michael et al, also in 2018 it was reported that the fixation step did not interfere with subsequent cross-linking.

Despite this potential lack of novelty, the manuscript is well written, and the used formaldehyde concentration is higher than used before. Hence, the described workflow is likely valuable for the community to improve their results. A revised manuscript might

therefore be suitable for publication. Supplementation of additional research data like usage of the described workflow to answer a biologically relevant question impossible to answer before would strengthen the impact of this work.

Please find some general comments and thoughts below:

- In the introduction the authors describe the issue of crosslinked species being low abundant, which is eventually problematic for data analysis. I suggest adding one or two sentences on affinity enrichment strategies, also used by the authors themselves, to alleviate this issue.
- The authors state that formaldehyde (FA) crosslinks (XL) are hard to detect by MS and assume they are formed at very low efficiency even at higher concentrations. I am wondering if FA links are maybe less stable than expected and if they are reversed prior to crosslinking even without heating? Can you comment on this? Also I was wondering if the authors searched for residual FA links (with or without reversing by heating) using known modification masses for FA (compare to work from Kalisman lab, where +12 and +24 Da are reported, <https://www.nature.com/articles/s41467-020-16935-w>)
- I am wondering why biotin-X-NHS did not pass the membrane and yielded 0 modified peptides within the live cells? It is not charged, seems reasonably hydrophobic and has a comparable size to crosslinkers like DSSO, DSBSO or PIR that are used for in vivo studies. Do the authors have a hypothesis why no modifications of biotin-x-nhs were found? Maybe the reaction time was simply too short (and hence shorter than ~1hrs one would do in a one-step XL-reaction)? Was the fragmentation site at the amide-bond considered for the search (which could improve scoring for modified peptides as fragment matching would be improved)?
- Supplemental Figure 1: Although this reviewer still has relatively young and good eyes, it seems not possible to see the cell-shape in the fluorescent/brightfield overlay - can the contrast be optimized to make it visible? Also, in the bright-field images the contrast seems suboptimal.
- Supplemental figure 2: Why does NHS ester labeling efficiency increase upon increasing FA concentration? This seems counterintuitive at first sight. Is it based on better permeabilization of membranes coming from the FA itself? If so, how does this bias the hypothesis that FA does not-lower NHS labelling efficiency (maybe it would indeed do so in fully permeabilized cells compared against each other). The latter assumption would be supported by supplemental figure 3A where permeabilization improves labelling efficiency.

Minor comments:

- Page 5: "...competition. but because..." Replace "." with ","
- Page 11: "Without permeabilization, we only observed a 1.2-fold increase in CSMs and

no change in the number of unique crosslinks (Fig. 4A)."

Seems confusing, change to something like

"Compared to live cells, with permeabilization we only observed a 1.2-fold increase in CSMs and no change in the number of unique crosslinks (Fig. 4A)."

July 29, 2024

Re: Resubmission of NCOMMS-24-22728-T

Dear Editor:

We thank you for the encouraging review of our manuscript. We have addressed the reviewers' comments by adding some additional data and clarifying our text where necessary. Below you will find the reviewer comments in blue text and our response in black text.

Reviewer #1

Overall, the authors do a thorough job of demonstrating this new workflow is effective at increasing in-situ XL-MS efficiency and expanding the crosslinkers available to do in-situ XL-MS with. The claims are well supported, the methodology is solid, and this work should be reproducible. There are a few minor issues outlined below, but once these are addressed this paper is ready for acceptance.

1. The authors should add a key to the data submitted to PRIDE. Even with a logical naming scheme the number of files make it challenging to match results to the experiments.

Agreed and our apologies. I too am often frustrated with opaque PRIDE entries. We have fixed it.

2. The authors were very thorough in explaining the indicators of cell disruption in the methods. If possible, they should more explicitly state the degree of disruption required to consider a cell distorted.

It is a little difficult to be fully quantitative, but we emphasized gross distortions, and calibrated our efforts by using two independent analysts. We have reworked the methods section slightly to emphasize that major distortions were mapped.

3. It may be useful to add a citation or rationale as to how the reverse formaldehyde cross-linking conditions were selected.

Agreed. We added a statement on page 19-20 that formaldehyde reversal was used to ensure effective digestion (it is a mild effect, near as we can tell). We also added a citation for the protocol (reference 44).

4. In Figures 2A and 2B, it would be helpful to say the number of replicates performed to get the error bars ideally in the figure caption.

Done.

5. In Figure 4A and 4B, the authors should add error bars and state the number of replicates performed. If it is only one replicate they should explicitly say so and acknowledge any limitations that may add to the analysis.

We performed *extensive* biological replicates using a simple shotgun method as readout, to set the conditions and stabilize the results. Once we finalized the method, we ran biological duplicates using the full fractionation scheme, but took the union of the results. The methods were updated to specify this precisely, to make both the strengths and weaknesses of the approach clear (page 24).

Reviewer #2

The remarkable result of this work is that cell fixation avoids the spatial distortion of proteins caused by prolonged exposure to the cross-linking reaction without preserving cellular structures in advance. In addition, formaldehyde fixation does not interfere with labeling and cross-linking reactions when excess formaldehyde is removed. The results could be of immediate interest.

1. However, the number of PPIs identified is not a very large number to move the field forward. Perhaps it would have been possible to show that the proposed approach has an undeniable advantage over the cross-linking of unfixed cells (by greatly improving the number and quality of PPIs or shortening the time of the cross-linking reaction...) if the hypothesis had also been validated with cross-linkers more suitable for an in-vivo/in-situ experiment, such as cleavable cross-linkers (DSSO, DSBU...), which reduce the search space and increase the identification of possible interactions.

This is an interesting comment, as we believe took pains to make a fair comparison that highlights the advantages of pre-fixation. The numbers of PPI's we generated are very high *when you consider the amount of sample input*, and quite enough to illustrate the advantage. Consider the following:

- It doesn't matter which crosslinkers we use, provided we don't pick obscure ones. We elected to stay with noncleavable crosslinkers, and we chose DSS as our control. DSS remains the most common crosslinking reagent, and it is widely regarded as being one of the better cell permeable crosslinking reagents.
- We do demonstrate a substantial increase over a basic, live-cell DSS method (see figure 4). Indeed, figure 4C clearly shows that the average quality of the hits we get are much superior to the live cells: the average STRING score increases quite a bit, which we would expect with better crosslinking data.
- We intentionally used a modest detection method: a 2D-LC method with light fractionation, using comparatively very little protein digest (about 5-10 micrograms injected, maximum, where most of the signal is linear free peptides). We added a statement to this effect on page 10 of the revised manuscript. Overall, the numbers we report are good for this level of analysis! Please also see points 2 and 4 below.

2. Cleavable cross-linkers would also have allowed a direct comparison of the results with recent studies already published in the literature where mainly cleavable cross-linkers were used.

As noted above, it isn't the goal to compare directly with recent studies. It is better to compare internally with an acceptable control. Cross-literature comparisons are fraught with danger at any rate and would cloud the study. It is difficult to exactly match analytical conditions (cell type, amount of lysate used, fractionation scheme, instrument, data analysis software). In many studies, important information is often left out and data analysis methods very opaque. For example, we have observed a body of work that over-inflates cross-link numbers based on a flawed analysis strategy. It is too early for cross-lab comparisons in this field.

We figured the best way forward was to use the most common *nominally* permeable crosslinker over the past 40 years, yet one that people have struggled to use effectively for deep *in situ* work. And as we have shown elsewhere, the success of cleavable crosslinkers really isn't that much different from non-cleavable crosslinkers.

3. While for enrichable crosslinking it would have been better to use a newer membrane-permeable crosslinking, tBu-PhoX, or to use both PhoX and tBu-PhoX to compare the results, evaluating the coverage improvement on the PPIs identifications when the fixing step is added.

We are puzzled by this comment. One goal of the work was to demonstrate that *permeabilization* (enabled by fixation) allows any crosslinker to be used, even non-permeable PhoX. Implementing tBu-PhoX is no different than using DSS: both are nominally membrane permeable. And we conclusively show we can make DSS better with fixation and permeabilization. A comparison between PhoX and tBu-PhoX would add very little to the study. You can use any crosslinker now. That is the point we illustrate and emphasize.

4. Therefore, expanding the crosslinkers used could have helped strengthen the hypothesis of the study by directly comparing PPI identifications for each crosslinker (permeable and non-permeable, cleavable and non...) when adding the cell fixation step before the crosslinking reaction, as well as for permeabilization.

We disagree and believe the points we raise above address this concern. It is unreasonable to require us to test all classes of crosslinker.

However, we did add a new set of data that we think elevates the paper. We did a 3X application of PhoX and generated some really impressive numbers – best in class when you consider how little starting material we used. We generated over unique 900 PPIs, more than double what we found with the 1X application of PhoX. We only used two small 10 cm plates of cells as input. This demonstrates the benefit of our method. We added text to page 13 and 14 and revised our Figure 5 to include the 3X PhoX data, showing approximately 300,000 CSMs and over 40,000 unique crosslinks. The interactome analysis was updated in this Figure, and the 1X PhoX data placed in supplementary, as a new Supplementary Figure 5. We are excited by these results. This should aid in addressing any lingering concerns from point 1 as well.

5. The conclusions are not straightforward to me. The results clearly show that if formaldehyde is removed prior to the addition of the crosslinker, it does not interfere with the crosslinking reaction. However, the results state that permeabilization after fixation improves the crosslinker yield, but the data show that the approach that provides more PPI and IntersXL identifications is that of 3 sequential shots, but this strategy was not used for PhoX, so it's not clear if more shots could have provided more PPI identifications. Therefore, which strategy is the best cannot be said with certainty.

We think the addition of the 3X PhoX data will help solidify the conclusions we draw. The numbers for both DSS and PhoX clearly tell the same story: multiple applications of crosslinker in a fixed environment (with permeabilization) adds considerably to the number and quality of the results. This isn't possible without fixation.

6. No lysine modification has been found after fixation, but have FA dimer crosslinkers been searched for (Nat Commun 11, 3128 (2020))?

We looked for these in separate experiments without FA reversal but found very little evidence for them. Bear in mind that we do a formaldehyde reversal step after workup, so we don't expect to see them in the normal approach.

7. It would be nice to add overlapping PPI analysis. Are all PPIs found in living cells included in Fix+Perm cells treated? If not, are those that differ in live cells due to disrupted cell structure, artifact, or are the 2 approaches complementary?

We elected to use a STRING analysis instead, which is more informative: a direct comparison with expected PPIs is a better metric of interactome integrity. Consider: at our depth of sampling, we can anticipate that a large fraction of the detected PPIs would have prior evidence of existence. You can see that is the case with 3X PhoX, where around 60% have the highest PPI STRING score. This number doesn't change from 1X PhoX, which is interesting. If our crosslinking method distorted interactome structure, we would expect multiple applications to make the ratio worse. Stability in this number indicates that repeat applications don't add distortion.

This is sensible. The cell is increasingly crosslinked and thus increasingly stabilized. We do note that the "Not Found" (N/F) fraction in the Figure 5 is about 20%. This likely contains (a) new interactions and (b) erroneous PPIs from poor FDR estimation. The latter is still a problem in the field at this scale of analysis, estimated here at 5% but possibly a bit higher.

8. Some specific mistakes have been identified:
 - a. Error in the text for the total reported XLs of PhoX: 168177  loop + intra + inter, it should have

been 53870 for comparing with 10722

b. 10722  intra + inter or 32084 for comparing with 168177.

We fixed our reporting in the text to be consistent throughout. For total unique crosslinks we enumerated all types: loop-links, intra- and inter-crosslinks. When breaking the numbers down further, we made sure to make the correct references. Thank you for catching this.

c. Figure 1 could be improved: the order of the bars should follow the order of the microscope images; while Supplementary Figure 3 would look more convincing if the same quantification as for Figure 1B were present.

Agreed. We have made the changes to Figure 1 and added the same analysis to Supplementary Figure 3 (which confirms our data in Figure 1).

d. Figure 5C shows redundant PPIs: the double lanes (edges) connecting the same 2 proteins (nodes) are duplicates. Therefore, fewer PPIs were found than declared.

This is a good point. We are recording AB and BA interactions as separate PPIs, as they are detected as such. The field is a bit inconsistent with such reporting. We adjusted the numbers, but they dropped only slightly.

e. Figure 5D & E also lacks a description of the crosslinker line colors.

Noted. The figure captions have been updated accordingly.

f. Figure 4C capture describe as red the bars for live cells, but it is more gold-ish.

Fixed, thank you.

g. Some inaccuracies and typos were found in the English, including the syntax; please review it.

We cleaned up some syntactical errors, but it's generally a solid piece of prose. Please let us know if there are blunders that should be corrected! The rest is creative license.

h. Some details about repeat injections are missing. No description of subsequent injections: sequential? Was there incubation between one and another? How long did the reaction last? Did the concentration of DMSO in the solution increase by more than 2% with multiple doses of crosslinker? Did this affect crosslinker penetration? Were the cells permeabilized before? This is not clear. It seems NO, so why was it stated that the permeabilization step is a necessary step to improve the in-situ crosslinking reaction? Why not apply it here, also with the multiple shot approach? In my opinion, a more in-depth analysis could have been done.

We added a short section in the methods section describing the process of multiple crosslinker applications (page 19). Yes, it is sequential application of the crosslinker (1x vs 3x). Basically, each application is followed by a short wash with vehicle-free buffer then a new aliquot of crosslinker is added. No additional permeabilization is required, as this persists. It is irreversible.

Reviewer #3

In their Study, Michael et al., demonstrate the advantage of a 2-step crosslinking approach using formaldehyde for fixation and stabilization of protein-protein interactions followed by membrane permeabilization and crosslinking with the amine-reactive crosslinkers DSS or PhoX. They show that fixation preserves the cellular ultrastructure while no competition to the amine-reactive crosslinkers formaldehyde was reported. This clearly improved their results over a classical one-step crosslink approach. Their proof of principle study in A549 cells shows nice proteome coverage and the idea of making more -also non membrane permeable - linkers accessible for "in vivo" crosslinking is great.

1. However, this reviewer is wondering if the idea is new per-se? Two step crosslinking approaches with formaldehyde to maintain cellular integrity were already reported in previous studies and, to give an example, were summarized in a review from the Huang lab back in 2018

(<https://www.ncbi.nlm.nih.gov/pmc/articles/PMC6022837/>). In that review applications to improve structural characterization but also for *in vivo* studies in combination with the affinity enrichable DSBSO linker are described. In line with the findings of Michael et al, also in 2018 it was reported that the fixation step did not interfere with subsequent cross-linking.

We also noted the prior use of formaldehyde crosslinking in the introduction to our paper and we cited the relevant studies. But what we did is different. The prior use involved very low concentrations (e.g., 0.025% FA), which is not enough for comprehensive cellular stabilization. All common cell stabilization protocols in immunochemistry use 1-4% FA. This is an important distinction! We suspect prior attempts pulled back on the percent FA because of fears that FA would compete with the crosslinker, which our study shows is not the case. It is the key to our approach. In addition, the previous work used a 2-step crosslinking procedure where the protein complexes of interest were enriched after mild FA stabilization, and then crosslinked on beads.

Further, we note that crosslinking with formaldehyde/glutaraldehyde has also been used for decades to support cryo-EM studies. What we show is that full-on cellular stabilization with formaldehyde is compatible with *in situ* crosslinking and enabling. We think that is novel and unexpected and have tweaked the abstract to emphasize the use of high concentration formaldehyde.

2. Despite this potential lack of novelty, the manuscript is well written, and the used formaldehyde concentration is higher than used before. Hence, the described workflow is likely valuable for the community to improve their results. A revised manuscript might therefore be suitable for publication. Supplementation of additional research data like usage of the described workflow to answer a biologically relevant question impossible to answer before would strengthen the impact of this work.

As we noted, the use of higher concentration is indeed the innovation. On the surface this may seem trivial, but it isn't. We encourage the reviewer to try it! You'll never go back.

We do note the detection of a wide range of PPIs and highlight an interesting one, probing its structure with AlphaFold Multimer but that is about as far as we need to go in a study of this nature. We are consistent with other advancements that report on new methods (including many in Nature Communications). A complete biological study will indeed be the subject of a separate paper, but it is important to have a dedicated manuscript on the methodology so it can be scrutinized (and used) by the community.

However, we did update our paper by adding the 3X PhoX data for more coverage of the interactome, and we updated Figure 5F. This figure uses CLAUDIO, a great tool that allows us to map all crosslinks on all of proteins and PPIs we detected (for which there is structure or AlphaFold data). It shows that all our detected crosslinks faithfully sample structure. We have updated the methods section on page 23 accordingly.

3. In the introduction the authors describe the issue of crosslinked species being low abundant, which is eventually problematic for data analysis. I suggest adding one or two sentences on affinity enrichment strategies, also used by the authors themselves, to alleviate this issue.

We refer to three papers in the introduction (references 11-13) that encompass a range of strategies (including affinity enrichment) to help alleviate the issue.

4. The authors state that formaldehyde (FA) crosslinks (XL) are hard to detect by MS and assume they are formed at very low efficiency even at higher concentrations. I am wondering if FA links are maybe less stable than expected and if they are reversed prior to crosslinking even without heating? Can you comment on this? Also I was wondering if the authors searched for residual FA links (with or without reversing by heating) using known modification masses for FA (compare to work from Kalisman lab, where +12 and +24 Da are reported, <https://www.nature.com/articles/s41467-020-16935-w>)

To the reviewer's first point, it is true that the FA chemistry is reversible, including perhaps the crosslinks. Our results suggest that many of the products of FA labeling (the amine-based ones at

least) are reversible, under the action of washing and perhaps via competition with the NHS-esters. It may be more precise to state that *irreversible* crosslinks are formed at very low efficiency. We've revised the text in our discussion to convey this thought. Whatever the reaction yield for true irreversible crosslinks, we don't see competition with NHS labeling. But we freely acknowledge the conundrum: FA stabilization is a requirement for preserving cellular structure throughout the process, which suggests some level of stability. A heating step is used to improve antigen recovery in immunochemistry, suggesting some stability in FA crosslinks. Quasi-stability may be enough, with the participation of other linkages as well.

To your second point, we did an open-modification search to see the range of possible reaction products after FA treatment. We did see some of the expected modifications, but these were very low in abundance (particularly compared to the NHS esters).

5. I am wondering why biotin-X-NHS did not pass the membrane and yielded 0 modified peptides within the live cells? It is not charged, seems reasonably hydrophobic and has a comparable size to crosslinkers like DSSO, DSBSO or PIR that are used for in vivo studies. Do the authors have a hypothesis why no modifications of biotin-x-nhs were found? Maybe the reaction time was simply too short (and hence shorter than ~1hrs one would do in a one-step XL-reaction)? Was the fragmentation site at the amide-bond considered for the search (which could improve scoring for modified peptides as fragment matching would be improved)?

We were somewhat surprised as well. We noticed that addition of the reagent led to a *dramatic* level of apoptosis, suggesting that the reagent is doing something! Strange, as it is very similar to sulfo-NHS-LC-biotin. (Perhaps there are undetected excipients in the stock?) However, a characteristic of apoptotic cells is the reduction of cytosolic pH (rather dramatically to around 6 or lower). This could be the answer: at this low pH our standard reaction time isn't long enough for efficient amide formation from an NHS ester. Plus, we have learned in this study that permeability spans quite a range; some of the reagent may get across but not very much. Admittedly we didn't look for reagent fragmentation but note that we had no trouble finding the labels in the fixed cell experiments (supplementary figure 3a).

6. Supplemental Figure 1: Although this reviewer still has relatively young and good eyes, it seems not possible to see the cell-shape in the fluorescent/brightfield overlay - can the contrast be optimized to make it visible? Also, in the bright-field images the contrast seems suboptimal.

Agreed! We have adjusted these images as much as possible. It's the best we can do without undue manipulation.

7. Supplemental figure 2: Why does NHS ester labeling efficiency increase upon increasing FA concentration? This seems counterintuitive at first sight. Is it based on better permeabilization of membranes coming from the FA itself? If so, how does this bias the hypothesis that FA does not-lower NHS labelling efficiency (maybe it would indeed do so in fully permeabilized cells compared against each other). The latter assumption would be supported by supplemental figure 3A where permeabilization improves labelling efficiency.

That is an interesting observation, especially for *E. coli*, where there is a slight uptick in labeling at 0.25% FA. We used the smallest neutral NHS reagent we could find to guard against significant changes in permeability, but we suppose it is possible. But notice that the trend for human cells is less obvious, and not large.

At any rate, if we understand the reviewer correctly, we are being asked to consider how enhanced permeability (by FA) might offset reduced reactivity of the NHS reagents (due to competition with FA). We don't think this is consistent with our findings. The results we get from the repeat applications of reagents continue to show there is a large pool of available amines for labeling, and thus competition (if it exists) is minimal.

8. Minor comments:

- a. Page 5: "...competition. but because..." Replace "." with ","

We rebuilt the sentence as it was clunky.

- b. Page 11: "Without permeabilization, we only observed a 1.2-fold increase in CSMs and no change in the number of unique crosslinks (Fig. 4A)." Seems confusing, change to something like "Compared to live cells, with permeabilization we only observed a 1.2-fold increase in CSMs and no change in the number of unique crosslinks (Fig. 4A)."

Yes, agreed. We improved the sentence.

We hope that the addition of the 3X PhoX data and the clarifying statements that we have added address all major concerns.

David Schriemer, Ph.D.
Dept. of Biochemistry and Molecular Biology
University of Calgary

Reviewers' Comments:

Reviewer #1

(Remarks to the Author)

This communication is significant, original, and well-supported. The methodology is sound and there is sufficient detail. All concerns brought up in the original review have been addressed and the paper ready for publication.

(Remarks to the Editor)

Reviewer #2

(Remarks to the Author)

As stated by the authors, the objective of this work is not to identify the optimal cross-linker for a specific experiment. Nevertheless, although there is a divergence of opinion regarding the selection of an appropriate cross-linker for in vivo reaction (like cleavable vs. uncleavable for in vivo experiments, DOI: 10.1021/acs.jproteome.0c00583, DOI: 10.1021/acs.chemrev.1c00786; or permeable vs not-permeable etc etc), the reported evidence demonstrates conclusively that this approach (fixation and permeabilization) will enhance the cross-linking reaction in vivo, regardless of the chosen cross-linker, while containing the spatial distortion.

This revised version of the paper incorporates several improvements, particularly the introduction of the 3x PhoX dataset, which serves to reinforce the main goal of this paper. Additionally, the various comments addressed provide further clarification of the points presented in this manuscript, thereby enhancing the overall quality of the paper. The purpose of the proposed overlap analysis of PPIs was to ascertain whether PPIs derived from different approaches represent complementary or redundant data sets. However, this can easily be done by consulting the tables included in the Supplementary Information, if of interest.

I concur that a FDR of 5% may be rather elevated. However, CSMs, URPs, and PPIs identified with a false discovery rate of 1% are duly included in the Supplementary Information table.

In my opinion the paper is suitable for publication, but I suggest few more minor revisions:

- There is a wrong indication about “supplementary Figure 5” at line 226 and 229, the text refers to figures 5A & 5B respectively. In the legend of figure 5 would be nice to indicate that also for 5D & 5E referred to 3x PhoX.

- In the table "Contents" of "Supplementary Information" is missing the description of Table S4.
- Line 228, the PPIs indicated for 3x DSS are 438 while in the Table S1 are reported 448 PPIs.
- Lines 74-76: "It too is a crosslinker, but it is quite different from the reagents used in XL-MS. Formaldehyde is very water soluble, is rapidly taken up into cells, and its reaction kinetics are fast." It could be rephrased to something like this: "It is also a cross-linking agent, but it is quite different from the reagents used in XL-MS. Formaldehyde is very water soluble, is rapidly uptaken into cells, and its reaction kinetics are fast."
- Lines 142-143: "These results are entirely dependent on the washing step, however." It may be better to use: "However, these results are entirely dependent on the washing step used."

(Remarks to the Editor)

Although I still disagree with some cross-linker choices, which are simply different opinions like which cross-linker to use (I would have used all permeable DSS and bu-PhoX or all non-permeable BS3 and PhoX), they clearly explain their decisions and make sense of it; they clarify that the focus of their work is on the method to optimize the XL-reaction without distorting the spatial proteome and not the choice of cross-linker.

I am glad that they have added the data set of the 3 shots of the PhoX cross-linker, taking up my suggestion to apply the best approach found for DSS to PhoX as well, thus clearly showing which is the best approach for all cross-linkers used (permeable and non-permeable).

In my opinion:

- "taken up into" should be substituted with "taken up by" or "incorporated into" or "absorbed into" or "uptaken into"

Reviewer #3

(Remarks to the Author)

The authors addressed all major concerns and significantly improved the quality of their manuscript. The addition of a 3xPhoX dataset nicely confirms their hypotheses initially made and further confirms the applicability of their protocol also for non-membrane permeable linkers.

I can now recommend this study for publication. Please find some minor comments below.

Minor comments:

- Figure 5: I'd suggest adding the statement fixed and or permeabilized to all bars in the bar chart of panel A, wherever applicable, to harmonize the design to Figure 4.

- To the comment of reviewer 3/question 5: I agree that the pH will have an influence on the reactivity of the NHS ester.

If I understood the rebuttal comments correctly, the authors were surprised by the fact that 0 modified peptides were detected in the live cells (assuming they agree with the reviewer's opinion to a certain permeability of biotin-x-NHS). They however added a contrasting statement on page 8 of the revised manuscript:

"We then chose biotin-X-NHS as a surrogate for a larger crosslinker and sulfo-NHS-LC-biotin as a surrogate for a charged crosslinker. Neither should be membrane permeable, and as expected, we detected no labeling of the proteome in the non-permeabilized cells for either reagent."

In line with the original comment of reviewer 3 it still seems surprising to me that biotin-x-nhs was not entering the cells/yielded no modifications at all. I would suggest to add the statement from the rebuttal explaining that biotin-x-nhs lead to significant apoptosis the text and as potential explanation instead, as this surprising behavior might indeed have influenced the result.

September 13, 2024

Re: Resubmission of NCOMMS-24-22728-T

Dear Editor:

We thank you for the second review of our manuscript. We have addressed all remaining concerns:

Reviewer #2

In my opinion the paper is suitable for publication, but I suggest few more minor revisions:

- There is a wrong indication about “supplementary Figure 5” at line 226 and 229, the text refers to figures 5A & 5B respectively. In the legend of figure 5 would be nice to indicate that also for 5D & 5E referred to 3x PhoX.

We have repaired this section.

- In the table “Contents” of “Supplementary Information” is missing the description of Table S4.

Added, thank you.

- Line 228, the PPIs indicated for 3x DSS are 438 while in the Table S1 are reported 448 PPIs.

Oops. The numbers in the paper are correct. We have updated Table S1 accordingly and reposted it.

- Lines 74-76: “It too is a crosslinker, but it is quite different from the reagents used in XL-MS. Formaldehyde is very water soluble, is rapidly taken up into cells, and its reaction kinetics are fast.” It could be rephrased to something like this: “It is also a cross-linking agent, but it is quite different from the reagents used in XL-MS. Formaldehyde is very water soluble, is rapidly uptaken into cells, and its reaction kinetics are fast.”

We prefer the flow and word selections of our text.

- Lines 142-143: “These results are entirely dependent on the washing step, however.” It may be better to use: “However, these results are entirely dependent on the washing step used.”

Our wording is clearer. We are referencing the washing step in general, rather than any particular washing step.

Reviewer #3

The authors addressed all major concerns and significantly improved the quality of their manuscript. The addition of a 3xPhoX dataset nicely confirms their hypotheses initially made and further confirms the applicability of their protocol also for non-membrane permeable linkers. I can now recommend this study for publication. Please find some minor comments below.

- Figure 5: I’d suggest adding the statement fixed and/or permeabilized to all bars in the bar-chart of panel A, wherever applicable, to harmonize the design to Figure 4.

Good idea, done. A new figure was uploaded.

- To the comment of reviewer 3/question 5: I agree that the pH will have an influence on the reactivity of the NHS ester. If I understood the rebuttal comments correctly, the authors were surprised by the fact that 0 modified peptides were detected in the live cells (assuming they agree with the reviewer's opinion to a certain permeability of biotin-x-NHS). They however added a contrasting statement on page 8 of the revised manuscript: "We then chose biotin-X-NHS as a surrogate for a larger crosslinker and sulfo-NHS-LC-biotin as a surrogate for a charged crosslinker. Neither should be membrane permeable, and as expected, we detected no labeling of the proteome in the non-permeabilized cells for either reagent."

In line with the original comment of reviewer 3 it still seems surprising to me that biotin-x-nhs was not entering the cells/yielded no modifications at all. I would suggest to add the statement from the rebuttal explaining that biotin-x-nhs lead to significant apoptosis the text and as potential explanation instead, as this surprising behavior might indeed have influenced the result.

Thank you for detecting the inconsistency. We adjusted the language in the text as follows:

"Neither reagent labeled the proteome of the non-permeabilized cells. (High levels of apoptosis were observed upon treatment, which can dramatically decrease the pH of the cytosol and reduce the labeling efficiency of NHS esters.)"

We have attended to all the editorial requests in the author checklist.

David Schriemer, Ph.D.
Dept. of Biochemistry and Molecular Biology
University of Calgary